# POLICY LEARNING FOR VIDEO STREAMING

## ABSTRACT

Facilitating good quality of experience (QoE) for Internet-based video services is a crucial real-world challenge. With remote/hybrid work, education, and telemedicine being here to stay, poor video quality adversely impacts the economy and society at large. The key algorithmic challenge in this context is *adaptive bitrate selection (ABR)*—continuously adjusting the video bitrate (resolution) to the prevailing traffic conditions. ABR algorithms struggle to maintain high resolutions while avoiding video stalls and long "lags behind live", and are the subject of extensive attention. In particular, ABR has, in recent years, been approached from different ML perspectives. However, disillusionment with applications of end-to-end deep reinforcement learning (DRL) to ABR have effectively led to abandoning policy learning for ABR altogether in favor of control-theoretic optimization methods. We demonstrate that, through more nuanced policy learning, substantial improvement over the state-of-the-art is achievable. Specifically, we show that applying deep-Q-learning to the output of a supervised predictive model bests alternative approaches. As we believe that the ABR domain is an exciting new playground for policy learning, we release our code for ABR policy learning and experimentation to facilitate further research.

## 1 INTRODUCTION

With hybrid/remote work and telemedicine becoming the new norm, providing acceptable quality of experience (QoE) for users when participating in video conferences, watching streaming video, etc., is crucial for the economy, and society at large. Yet, despite massive investment in communications infrastructure, severe service quality problems persist, including stalls and long "lags behind live" for real-time services like video conferencing, and frequent re-buffering and low resolutions for video streaming[1]. The main frustrations and pains are the share of users living in rural areas and lower GDP countries, as well as users in very densely populated areas, where many users compete over the same network bandwidth. Facilitating good video QoE across geographical and political boundaries is therefore also important for bridging the "digital divide".

Consider, for instance, a Zoom conversation between two distant parties, or a Netflix movie streamed from a Netflix video cache to a user's TV. In both scenarios, the service provider's (Zoom/Netflix) content traverses third-party networks (e.g., the user's ISP, say, AT&T or Telefonica). Hence, the video content must traverse various network segments over which the service provider has little, if any, control. Delivering the data traffic thus involves competing over the limited network bandwidth with other services and users sharing the same infrastructure. Moreover, some of the network segments (e.g., mobile and cellular networks) exhibit notorious bandwidth volatility, with the available bandwidth changing from one moment to the next.

In light of the above, a crucial challenge for attaining high video QoE is modulating the data transmission; injecting video traffic into the network "too quickly" at a certain point in time might overwhelm the network, resulting in video stalls/rebuffering due to data being lost or delayed; injecting video traffic too slowly might be insufficient for supporting high video quality (e.g., HD). To address this challenge, video services employ online *adaptive bitrate selection* (ABR). Under ABR video streaming, the video *bitrate* (resolution) at which video content is delivered is continuously adapted to what the network can accommodate at any given time.

---

[1]In the US, even before the rise in Internet traffic following the Covid pandemic, Netflix and Amazon subscribers with premium-tier Internet access ($\geq$ 250Mbps) enjoyed video in HD less than $40\%$ of the time on average (Schmitt et al., 2019). Mass video streaming in HD, let alone $4K$, remains a distant dream.

Due to its popularity and importance, ABR-based video streaming has been the subject of extensive attention from both researchers and practitioners (Huang et al., 2015; Jiang et al., 2014; Yin et al., 2015; Sun et al., 2016; Mao et al., 2017; Akhtar et al., 2018; Spiteri et al., 2016; 2018; Yan et al.). In particular, various ML-based approaches to ABR algorithm design have been proposed. A seminal, highly influential such proposal (namely, Pensieve (Mao et al., 2017)) involved applying end-to-end deep reinforcement learning (DRL) to ABR. Attempts to realize this approach in the wild have, however, proven unsuccessful. As highlighted by Yan et al., DRL's notoriously brittle optimization process resulted in DRL policies that fare well in training performing poorly in deployment. Following the disillusionment with DRL-based ABR policy learning, policy learning for this task has been abandoned altogether. Instead, state-of-the-art ABR algorithms apply *model predictive control* (MPC) for *online* optimization of QoE with respect to a *finite time horizon* (Yin et al., 2015; Sun et al., 2016; Akhtar et al., 2018; Yan et al.). MPC suffers from two well-studied drawbacks (Lowrey et al., 2019; Bhardwaj et al., 2021). First, uncertainties regarding the model can naturally yield bad decisions. Second, MPC optimization occurs *online*, and so setting the look-ahead window to be too far into the future can be associated with prohibitively expensive computational burden and time.

In this study, we revisit ABR policy learning. We present a simple ABR scheme, called "Wolfi"[2]. Wolfi applies deep-Q-learning to the exact same inputs required for MPC's online optimization process, thus replacing MPC optimization with policy learning. By employing Q-learning, Wolfi is capable of *learning* from interaction with the environments how to contend with uncertainties in MPC's predictive model, and also to incorporate longer time horizons into its decision making process, thus addressing MPC's two main deficiencies. We contrast Wolfi's achieved video QoE with the state-of-the-art, including different flavors of MPC proposed for ABR. Our evaluation framework involves streaming actual video across an emulated network environment and leverages real-world, empirically-derived traffic traces. We show that Wolfi consistently and robustly outperforms alternative schemes across different datasets and QoE metrics.

We view ABR as an exciting new playground for policy learning, for multiple reasons. First, even though the problem may appear simple at first glance – the control algorithm only has to output a uni-dimensional value (the video bitrate) at each point in time – ABR has emerged as an important and persistent challenge. ABR algorithms have to deal with network environments that vary greatly in size, bandwidth, latency, level of competition, and more. The daunting breadth of possible network environments in which an ABR protocol might operate, and the typical dynamism of the network environment, pose significant challenges to ABR. Second, ABR crucially impacts user experience for services like video streaming and video conferencing, which constitute the dominant share of Internet traffic. Better ABR algorithms are key to enabling more people to work and study from home, better telemedicine, and more. To support further research on policy learning for video streaming, we release our code for policy learning and experimentation.

## 2 ADAPTIVE VIDEO STREAMING: BACKGROUND AND CHALLENGES

### 2.1 BACKGROUND: ADAPTIVE BITRATE (ABR) SELECTION

We focus on HTTP-based video streaming, the primary method for delivering video traffic across the Internet, used, e.g., for video-on-demand (VoD) and live streaming. The video is divided into *chunks* of fixed duration $L$ (e.g., 2 seconds). Each chunk is encoded at $M$ video *bitrates*, e.g., SD, HD, etc. A higher bitrate corresponds to the video chunk being encoded in higher resolution, and so requires more data (in bits). Video chunks are transmitted from the video *server* to the video *client*. The bitrate at which each video chunk is transmitted is dictated by the ABR algorithm. Video chunks are delivered to the video client in FIFO order. Only after a video chunk is *fully* downloaded, does the video client start playing the video chunk to the viewer.

Generally speaking, the input to the ABR algorithm consists of two components: (1) statistics relating to the downloads of recently transmitted video chunks, and (2) the current occupancy of the *video buffer* at the client, i.e., the duration of locally stored video (in seconds) at the video client. The output of the ABR algorithm is the bitrate of the next video chunk to be transmitted. Intuitively, the faster the anticipated download speed of the next video chunk, and the more seconds of video stored locally at the video client, the more risk tolerant the ABR algorithm can be. Suppose, for

---

[2]Wolfi is the world's smallest octopus.

instance, that the local video buffer at the video client only contains 1 second of stored video. In this scenario, if the download time for the next video chunk exceeds 1 second, the video client will experience video *stall/rebuffering*.

ABR algorithms must strike a delicate balance between two desiderata: maximizing video quality and minimizing video rebuffering events and duration. The higher the chosen video bitrates, the higher the risk of video rebuffering, since higher bitrates require more data to be transmitted across the network. Low bitrates, however, translate to the viewer watching the video in low resolution.

A standard metric for quantifying quality of experience (QoE) for video streaming is the following. Let $K_i$ denote the $i$'th chunk of the video. Let $Q(K_i)$ be the video quality at which video chunk $K_i$ was downloaded. Video quality can be measured simply in terms of the video bitrate (as in (Mao et al., 2017)), or via more nuanced measures like the structural similarity index measure (SSIM)—a method for predicting the viewer-perceived quality of digital images and videos (see (Yan et al.)). Let $T(K_i)$ be the total transmission time of chunk $K_i$, and let $B_i$ be the occupancy of the video buffer when chunk $K_i$'s transmission started. The QoE for chunk $K_i$ is defined as:

$$QoE(K_i, K_{i-1}) = Q(K_i) - \lambda|Q(K_i) - Q(K_{i-1})| - \mu \cdot max\{T(K_i) - B_i, 0\} \qquad (1)$$

Observe that $max\{T(K_i) - B_i\}$ captures the rebuffering/stall time associated with downloading chunk $K_i$. $\lambda$ and $\mu$ are coefficients that determine the tradeoff between rewarding video quality and penalizing bitrate "jitter" and rebuffering time. The QoE for an entire video is defined as the average QoE across all its chunks.

**ABR as sequential decision making.** ABR can naturally be cast as a sequential decision making task in which an agent interacts with its environment over discrete time steps and the objective is to maximize the agent's cumulative reward (Mao et al., 2017). Here, the agent is the ABR algorithm[3], and the environment represents the communication network interconnecting the video server and the video client. At each time step $t = 1, \ldots$ the agent observes a state of the environment $s_t$ from a (possibly infinite) set of possible states $S_t$, where $s_t$ can incorporate statistics regarding previously downloaded video chunks and the current occupancy of the video buffer, and selects an action $a_t$ from action set $A$, with $A$ corresponding to fixed set of video bitrates ($|A| = M$). After selecting action $a_t$, the agent observes reward $r_t = QoE(K_t, K_{t-1})$. The agent's goal is to optimize the cumulative reward $\Sigma_{t=1}^{T} r_t$, where $T$ is the number of video chunks in the video. An ABR algorithm translates to a decision making policy $\pi : S \rightarrow \Delta(A)$, where $\Delta(A)$ is the space of probability distributions over the set of actions (bitrates) $A$. ABR algorithms differ in their observations about the environment (which constitute the state) and in how states are mapped to bitrates.

## 2.2 Handcrafted, Control-Theoretic, and ML-Based ABR Algorithms

Over the years, many ABR algorithms have been proposed. We cover important classes of such algorithms below.

**Classical ABR algorithms.** The two main considerations for decision making are the future bitrate and the buffer occupancy. Simple but effective early methods use either the bitrate only (Jiang et al., 2014) or buffer occupancy only (Huang et al., 2015) for selecting bitrates. This, however, can yield suboptimal decisions. Model Predictive Control (MPC) uses both via a two step procedure:

1. Predict the download times for the next $k$ video chunks (for some fixed $k > 0$), at all possible bitrates, across a bounded look-ahead future horizon

2. Solve an exact optimization problem to determine the sequence of bitrates that maximizes a QoE objective (of the above form) with respect to the predicted download times.

Various flavors of MPC-based ABR algorithms, differing in the predictive model used (Yin et al., 2015; Sun et al., 2016; Yan et al.) and the robustness of the optimization process (Yin et al., 2015; Yan et al.), have been proposed.

**Machine learning methods for ABR.** Recently proposed methods use machine learning for improving ABR. Multiple methods, including Fugu (Yan et al.), employ MPC with respect to predicted download times of the upcoming video chunks. Fugu uses a deep neural network prediction

---

[3]Traditionally, ABR algorithms are executed at the video client, which requests the next video chunk at its chosen bitrate from the video player.

model,the Transmission Time Predictor (TTP), to derive probabilistic forecasts of future download times. Stochastic MPC is then used to optimize QoE with respect to these forecasts. In another thrust, Pensieve employs *end-to-end deep RL* for ABR policy learning (Mao et al., 2017). However, while Mao et al. (2017) demonstrated better performance than previous methods, subsequent large-scale empirical investigation demonstrated that, when unleashed in the real world, Pensieve exhibits poor QoE (Yan et al.). This can be attributed to the gap between Pensieve's training environment (a simulator) and the complexity of real-world traffic conditions, as well as to DRL's notoriously brittle optimization process.

# 3 REVISITING ABR POLICY LEARNING

## 3.1 INTRODUCING WOLFI

Prior attempts at policy learning for ABR were hampered by end-to-end DRL's notorious high sample complexity and sensitivity to hyperparameter tuning. Today's state-of-the-art, as embodied by Fugu (Yan et al.), in contrast, predicts network behavior from raw data via supervised learning, and then executes an optimization-based policy (MPC). This, however, might fail to take advantage of specific regularities in the environment to inform the decision policy. We propose an alternative approach, Wolfi, a new ABR algorithm. We show that by replacing MPC optimization with policy learning *on the exact same inputs*, Wolfi overcomes MPC's weaknesses.

**Wolfi's input:**

1. Current buffer occupancy level

2. video quality (as quantified by SSIM) of the last video chunk

3. video quality for each of the next $H$ video chunks, $K_{t+1}, \ldots, K_{t+H}$, at each of the possible $M$ bitrates

4. Expected download time (a real number) for each of the next $H$ video chunks, $K_{t+1}, \ldots, K_{t+H}$, at each of the possible $M$ bitrates

To calculate the expected running time for a video chunk at a certain bitrate, Wolfi uses the Transmission Time Predictor (TTP), introduced by Yan et al.. For any future video chunk $K_{t+1}, \ldots, K_{t+H}$ and video bitrate, TTP outputs a probability distribution over 21 bins, representing ranges of download times: $[0, 0.25), [0.25, 0.75), [0.75, 1.25), \ldots, [9.75, \infty)$. Wolfi's input download time values are derived from TTP's output by taking the expectation over all bins' median values (using a high real value for the last TTP bin).

**Wolfi's output** is a probability distribution over bitrates, from which the bitrate for the next video chunk is drawn.

**Wolfi's policy** is implemented using a Deep Q-Network (DQN). DQN optimizes a state-action Q-function of the form $Q^\pi(s, a) = E[R_t | s_t = s, a_t = a]$, where $R_t = \Sigma_{k=0}^\infty \gamma^k r_t + k$ the cumulative *discounted* reward (the return) after taking action $a$ at state $s$ for discount factor $\gamma \in [0, 1)$. As discussed in Section 2.1, in our context, actions translate to bitrates and rewards to QoE scores. Our implementation utilizes a Dueling DQN achitecture. Dueling DQN separates the learning of state-value and advantage functions. The output of each of the two separate streams is combined to produce an estimate of the Q-function value by an aggregating layer. This separation has been shown to accommodate more effective learning, resulting in faster convergence times and better overall performance.

**Implementing Wolfi.** In our implementation, we adopt the TTP architecture from (Yan et al.). A different instance of TTP is trained with respect to each of the next $H = 5$ chunks, $K_{t+1}, \ldots, K_{t+H}$, with the input to each TTP instance comprising statistics pertaining to the most recently downloaded 8 video chunks and the size of the future video chunk. The DNN realizing TTP is fully connected, with two hidden layers, each consisting of 64 neurons each. Wolfi's DQN consists of a single hidden layer. In our evaluation, the number of bitrates is 10 and $H = 5$, and so Wolfi's DNN's input and output layers are of sizes 102 and 10, respectively. Wolfi's hidden layer contains 256 neurons.

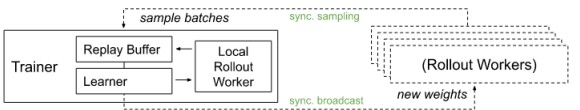

Figure 1: Ray's DQN architecture

## 3.2 TRAINING WOLFI

Training Wolfi consists of two phases: first training TTP, and then training the DQN.

**Training the predictor (TTP).** TTP is trained on historical downloads data of video chunks from the *deployment environment*. TTP is trained using standard supervised learning techniques (stochastic gradient descent), with the objective of minimizing the cross-entropy loss with respect to TTP's output probability distributions and the actual (discretized) download times. See (Yan et al.) for additional details regarding TTP's training process.

**Training Wolfi's policy.** To train Wolfi, we used to Ray framework (Moritz et al., 2018). Our training process is as follows. We repeatedly collect data from streaming a video to *multiple* video clients *in parallel* (15 in our evaluation) over a stretch of time lasting several minutes ($9 - 10$ in our evaluation). Since ABR decisions typically occur at the granularity of seconds, this amounts to thousands of data points per such iteration. To avoid strong correlation between data points, we use an experience replay buffer $D$. After collecting the data points, mini-batches of a fixed size (32 in our evaluation) are repeatedly sampled and used to update the DQN link weights. To facilitate exploration, we use the $\epsilon$-greedy method, sampling actions (bitrates) from Wolfi's output with probability $1 - \epsilon$ and from the uniform distribution over actions with probability $\epsilon$. The cumulative rewards (the returns) for actions are calculated for the appropriate choice of discount factor $\gamma$ ($\gamma = 0.8$ in our evaluation) and future horizon (the 15 next actions in our evaluation).

## 3.3 WHY WOLFI?

Prior attempts at policy learning for ABR, namely, Pensieve, employed DRL to learn end-to-end mappings from "raw" input features to video bitrates. These attempts were hampered by DRL's notorious high sample complexity and sensitivity to hyperparameter tuning. Today's state-of-the-art, in contrast, employs MPC with respect to a predictive model of future download times and a finite time horizon. Two main predicaments of MPC optimization are (1) sensitivity to uncertainty in the predictive model, and (2) the time horizon being too narrow due to scalability limitations of MPC's *online* optimization.

We attribute Wolfi's success (see Section 4.1) to its application of (deep-)Q-learning to features consisting of the *exact same* information used for MPC optimization (future download times, SSIM information, and current buffer size occupancy). In doing so, Wolfi overcomes the two above discussed limitations. First, by learning a decision policy through interaction with the environment, Q-learning can *learn* how to compensate for inaccuracies in the outputs of the predictive model. Second, by performing policy learning *offline*, Q-learning can consider longer time horizons (15 future video chunk downloads in our evaluation). By using TTP's output as the input to its decision making, Wolfi also benefits from the careful, domain-specific engineering that went into generating TTP. Consequently, Wolfi can learn more concisely represented, and better-generalizing policies than those learned by end-to-end DRL (which must extract features from raw observations).

## 4 EVALUATION

## 4.1 EVALUATION FRAMEWORK

We built on the experimental testbed in Puffer (Yan et al.) to create a framework for ABR policy training and evaluation. In the Puffer testbed, a video server streams video to multiple video clients. The network conditions between the server and each of the clients are determined by a network emulator that takes as input traces that specify how the available network bandwidth should change across time and limits the available capacity accordingly. In our training and evaluation, we used

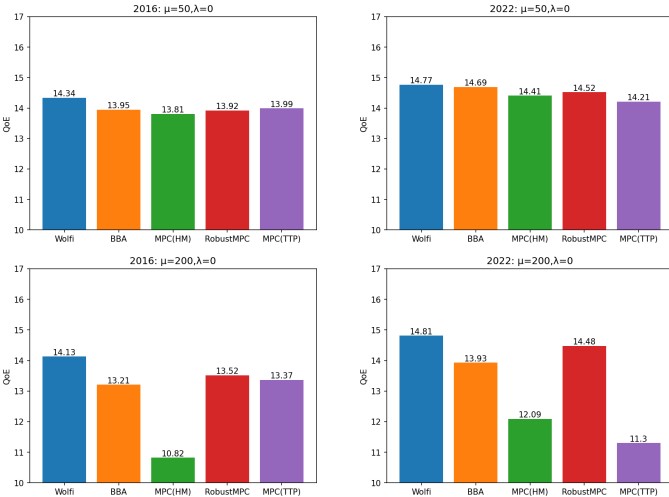

Figure 2: Performance comparison under a QoE metric which is weakly sensitive ($\mu = 50$, top row) and highly sensitive ($\mu = 200$, bottom row) to rebuffering. In both case, the metric is insensitive to jitter ($\lambda = 0$). Results are shown for datasets 2016 and 2022. Wolfi dominates in all cases. For high buffering sensitivity, RobustMPC is the strongest baseline, while the naive baseline BBA is the strongest for low buffering sensitivity.

empirically-derived traces from real-world networks, as shall be discussed below, to limit the bandwidth between the video server and the video client (the "downlink"). The bandwidth between each video client and video server (the "uplink") was fixed to be 12Mbps (to avoid client-to-server communication being the communication bottleneck) The (base) end-to-end latency between the server and each client was set to be 40ms.

Our policy training framework utilizes Ray (Moritz et al., 2018). Each video client is associated with a dedicated "rollout worker", which stores a DNN (DQN for Wolfi) with the current link weights. For every video chunk downloaded by a client, the server communicates with the appropriate rollout worker to send it the relevant input features and receive the derived bitrate. The server then instructs the client to download the next chunk at that bitrate. The client-specific rollout workers are tasked with computing the returns for each data point and storing the data (e.g., in a shared replay buffer, as discussed in Section 3.2). After sampling data mini-batches from the stored data, and using these to update the link weights of the DNN, the new weights are propagated to all the rollout workers, and the process continues. This procedure illustrated in Figure 1.

## 4.2 EVALUATION METHODOLOGY

**Video:** We use 10-minute long prerecorded videos with chunk duration of 2 seconds. The chunks are encoded in 10 different H.264 versions ranged from 240p60 video with a constant rate factor (CRF) of 26 to 1080p60 with a CRF of 20.

**Network dataset:** We used the FCC broadband America dataset from August 2016, June 2018, and June 2022. For each year we created a corpus by randomly selecting $500 - 1000$ from the "Web Browsing" category (as in (Mao et al., 2017). To ensure that the network poses nontrivial challenge to ABR algorithm, we only considered traces that met the following criteria: (1) average throughput of less than 6Mbps, (2) minimum throughput above 0.2Mbps, (3) median throughput less than 3Mbps, and (4) the trace lasting at least 10 minutes (the specific numbers are affected by the encoding of the prerecorded video). We used a random sample of $80\%$ of our corpus as the training set and $20\%$ as the test set.

**QoE metric:** We used the QoE metric defined in Section 2.1. We experimented with different combinations of assignment of values to the rebuffering penalty ($\mu$) and the bitrate jitter penalty ($\lambda$). Specifically, we set $\mu = 50, 100, 200$ and $\lambda = 0, 0.5, 1$.

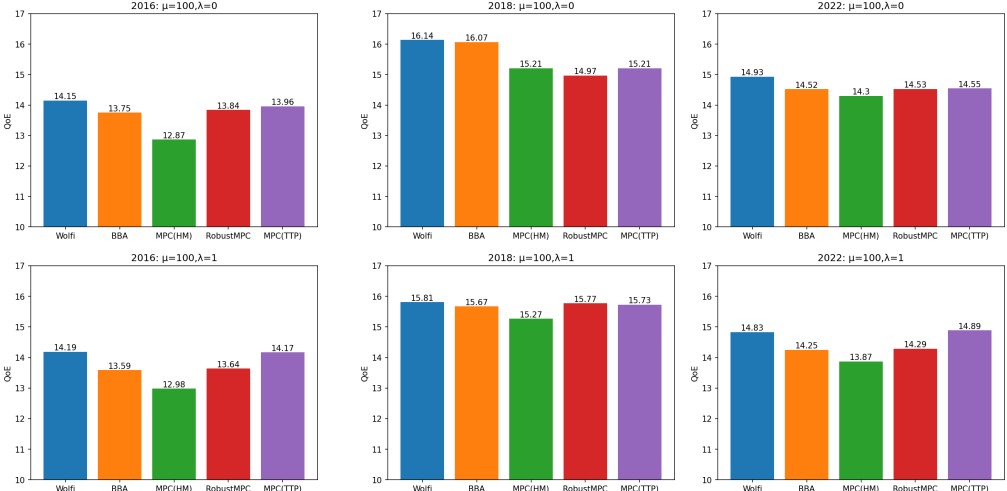

Figure 3: Comparison with a QoE metric with intermediate sensitivity to buffering ($\mu = 100$). The top row shows results for insensitivity to jitter ($\lambda = 0$), and the bottom row is for jitter penalty $\lambda = 1$. Results shown for the three datasets (2016, 2018, and 2022). Wolfi dominates in all cases. The simple baseline BBA often dominates all the baselines.

**Evaluated protocols:** We evaluated 6 ABR algorithms: (1) **BBA**, (2) **MPC(HM)**, (3) **RobustMPC**, (4) **MPC(TTP)**, (5) **DRL**, and (6) **Wolfi**. Algorithms 1-4 constitute the state of the art (see (Yan et al.)) and include a buffer-based scheme (BBA), shown to perform surprprisingly well in the wild, and three flavors of MPC optimization that have been applied to ABR and evaluated extensively: MPC(HM) (Yin et al., 2015) applies MPC optimization to a predictive model that uses the harmonic mean of the download speed of the 5 most chunk downloads as an estimate for the download speeds for the upcoming chunks. RobustMPC (Yin et al., 2015) is an extension of MPC-HM that optimizes worst-case QoE with with respect to prediction error margins. MPC(TTP) (called Fugu in (Yan et al.)) applies a *stochastic* MPC optimization process with respect to a predictive model informed by TTP. DRL is the deep reinforcement learning ABR algorithm from (Mao et al., 2017) (called Pensieve). We use the implementations of BBA, MPC(HM), RobustMPC, and MPC(TTP) from (Yan et al.).

For each combination of dataset and QoE metric, we trained a Wolfi policy by first training TTP on the data and then training Wolfi. MPC(TTP) also used the TTP trained on the particular dataset. QoE scores for different ABR algorithms are averaged over several runs.

### 4.3   EVALUATION RESULTS

Overall, we contrasted Wolfi's induced QoE with that achieved by the other ABR algorithms across 27 environments (3 datasets × 9 value assignments to the QoE metric coefficients). Figures 4.1 and 3 present a representative subset of results (encompassing 10 of these environments).

**Wolfi robustly outperforms the other ABR algorithms.** Observe that each of the different evaluated flavors of MPC, and the handcrafted buffer-based schemes (BBA) shines in different settings (dataset and QoE metric combination). Wolfi, in contrast, robustly and consistently outperforms the other ABR algorithms. We note that even a seemingly small change in average QoE can have important implications for overall user experience. For instance, in one of the runs for the 2022 data and $\mu = 100$, Wolfi's QoE score was $14.92$ and MCP(TTP)'s was $14.35$. A closer look at the factors contributing to this QoE gap revealed that Wolfi induced a rebuffering event on $9\%$ of the 100 empirical traces used as test, whereas MPC(TTP) caused rebuffering for $38\%$ (!) of traces. Since each trace represents a 10-minute video download, this translates to many more video viewers experiencing rebuffering. The total absolute rebuffering time for MPC(TTP) was also substantially higher on this run. As in the empirical analysis in (Yan et al.), our findings also reveal that the DRL

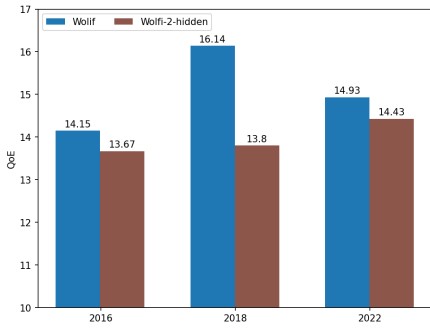

Figure 4: Adding an extra hidden layer to Wolfi's architecture reduces performance.

benchmark exhibits consistently worse QoE than the other benchmarks. We therefore remove DRL from consideration in the analysis below.

**Uncertainties relating to MPC's predictive model yield poor QoE and high variability in QoE. Wolfi and RobustMPC address such uncertainties.** Notably, when $\mu = 200$ (Figure 4.1), MPC(HM) and MPC(TTP) exhibit much worse performance than RobustMPC. A deeper dive into the results suggests why: the rebuffering time induced by both MPC(HM) and MPC(TTP) on the test data is much higher than that induced by Wolfi and RobustMPC (382s and 472s *vs.* 13s and 6s). We attribute this to the fact that MPC(HM) and MPC(TTP) optimize QoE with respect to predicted future download times without taking into account prediction uncertainty. Suppose, for instance, that the current buffer occupancy at the video client is 2s and the forecast is that the next video chunk can be downloaded at the highest bitrate in 1.95 seconds. Maximizing QoE while disregarding the risk of prediction error would lead to the highest bitrate being chosen. If, however, the actual download time is 2.1s, the video client would experience 0.1s of rebuffering. While this risk can manifest for any choice of value assignments to the QoE metric, when the rebuffering penalty $\mu$ is high (200), the cost of false decisions is dire. Wolfi and RobustMPC content with prediction uncertainty in different ways: RobustMPC optimizes worst-case QoE with respect to a prediction error model, whereas Wolfi *learns* how to best incorporate TTP's predictions into its decision making.

The QoE scores achieved by BBA, RobustMPC, and Wolfi, are fairly constant across different runs for the same choices of dataset and QoE metric. MPC(HM) and MPC(TTP), in contrast, exhibit high variability across runs. Once again, this can be attributed to the susceptibility of both algorithms to prediction errors, and to the nondeterministic nature of the environment. RobustMPC and Wolfi address prediction uncertainty, whereas BBA only makes decisions based on the buffer occupancy and makes to assumptions about future download times at all.

**TTP-informed predictive modeling yields better QoE.** Observe that MPC(HM) is almost always dominated by MPC(TTP). This is to be expected since the supervised predictive model of MPC(TTP), which relies on extensive historical information about the deployment environment, is superior to the simple online harmonic-mean-based predictions used by MPC(HM).

**Handcrafted policies are surprisingly robust but cannot to adapt to the QoE metric.** As also highlighted in (Yan et al.), the manually crafted, simple buffer-based heuristic (BBA) is surprisingly hard to beat. A closer examination of BBA's behavior in our experiments, however, reveals that it consistently prioritizes higher video quality (as measured by SSIM) over minimizing rebuffering time. Indeed, BBA induces consistently higher rebuffering times than both Wolfi and Robust-MPC. When the QoE objective places a high weight on avoiding rebuffering (e.g., when setting $\mu = 200$), this behavior by BBA can come at a hefty performance price.

**Wolfi's longer look-ahead window is conducive for ABR decision making. Smaller DNNs are sufficient.** To understand why Wolfi succeeds in providing better QoE, we conducted various additional experiments where we varied Wolfi's look-ahead horizon (15), discount factor ($\gamma = 0.8$) and DNN architecture (which consists of one hidden layer). Our findings suggest that a shorter look-ahead horizon yields worse QoE, while a longer look-ahead window renders learning less efficient while not leading to any performance gains. This suggests that Wolfi's ability to look further ahead into the future than MPC (due to MPC's optimization occurring online) is indeed beneficial for de-

cision making. Figure 4.3 contrasts the QoE achieved by Wolfi with the DNN architecture discussed in Section 3.1 and with a DNN architecture that contains a second hidden layer with $64$ neurons. The QoE metric used is with $\lambda = 0$ and $\mu = 100$. As seen in the figure, Wolfi with the smaller DNN architecture achieves better performance. In general, using semantically rich features as the input to Wolfi's policy, instead of mapping from actions to raw data, enables learning good policies using modest-size DNNs.

### 4.4 Main Takeaways

Our findings establish the following:

- Wolfi is robustly superior, in terms of QoE, to the other evaluated ABR algorithms, including the three flavors of MPC-based ABR algorithms.

- Wolfi's performance gains are due to its ability to better contend with uncertainty in the predictive model than MPC optimization and to its ability to optimize QoE with respect to a longer time horizon than MPC.

## 5 Discussion and Future Research

Our focus here was on exactly matching the inputs and output of MPC methods for the purpose of contrasting MPC optimization with policy learning. While we showed that Wolfi indeed outperforms MPC, this only sets a lower bound on the performance gains achievable via policy learning. In particular, the following are interesting directions for future research.

**Alternative features.** We believe that better input features for policy learning exist. For instance, a natural first step would be using as input the activations of the penultimate layer of the TTP, rather than its outputs (which are not as well-suited as inputs to neural networks).

**Improving TTP.** TTP attempts to predict the download times for the next $5$ chunks. The download time of future video chunk, however, involves nontrivial dependencies on the ABR algorithms previous actions (choices of bitrates), and is therefore hard to accurately predict. Instead, the future network throughput impacting the next $5$ chunks could be directly predicted, thus avoiding dependencies between predictions and future actions. Furthermore, the distribution of the download times can be described by simple parametric functions, rather than non-parametric histograms. We leave this investigation for future work.

**Hybrid methods.** Another interesting direction for future research is devising hybrid approaches for combining Q-learning with MPC. The limitations of MPC have led to various attempts to combine model-free reinforcement learning with MPC. One approach is contending with MPC's finite horizon by learning a value function that captures the cumulative reward after the MPC horizon (Lowrey et al., 2019). Another approach is "blending" MPC with Q-learning within the finite-time horizon (Bhardwaj et al., 2021). Investigating to what extent different combinations of MPC and Q-learning (and, more broadly, model-free RL) can be useful for further improving QoE is an intriguing question.

**Limitations of our approach.** Like all methods, ours also has limitations. Those include: (1) reliance on TTP-informed predictive model, whose predictions can degrade when network conditions differ from the training distributions, and (2) lack of theoretical performance guarantees (a limitation shared with most deep learning methods).

## 6 Conclusion

We revisited policy learning for ABR, showing that applying deep Q-learning to the outputs of a supervised predictive model bests the state-of-the-art, including various flavors of MPC optimizations. We view our results as but scratching the surface of policy learning in this important and timely context. We regard the ABR domain as an exciting new playground for policy learning and have released our code to facilitate further research in this area.

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
