# OpenReview forum: "Policy Learning For Video Streaming"
_ICLR.cc/2024/Conference — Submitted to ICLR 2024_

### Official Review · Reviewer_9tk1 · 2023-10-19

**Soundness:** 2 fair
**Presentation:** 2 fair
**Contribution:** 1 poor
**Rating:** 3
**Confidence:** 4

**Summary:**

This paper proposes a RL-based policy optimization for ABR video streaming. Finding optimal bit-rate selection is a old problem and yet, it has been shown that the state-of-the-art algorithm does not work quite well for diverse set of network traces. This paper developed a DQN-based approach and has shown that it can outperform the existing MPC-based approaches.

**Strengths:**

1. The algorithm works well compared to other MPC algorithms on Puffer.

2. The rationale behind the proposed algorithm is clear.

**Weaknesses:**

1. The main weakness seems to be the lack of novelty. Of course, DQN has not been used before, however, RL approach has been used (Pensieve). The paper did not explain why their approach has a better performance compared to Pensieve. The reward structure is the same, hence, the reviewer is wondering what are the advantages.

2. Even though the performance seems to be good, but it is not improving by a lot. While it is fine in general, but, given the fact that the methodology is not novel and the improvement is not a lot, my recommendation will be towards rejection.

3. The proposed algorithm has not been tested for a wide range of dataset in particular for 5G dataset.

**Questions:**

1. Can the authors have some comparison with Pensieve?

---

> ### Author Response · Authors · 2023-11-20
>
> Thanks for your review,
> 1. Our method is the first one, to our knowledge, to succeed in this complex task. Pensieve failed to do so, and Puffer show it in their paper.
> 2. Altough it may seem the performance does not improves a lot, for network purposes this actullay means a lot - every sub QoE is hard to improve because in the streaming environment there are many types of clients and the algorithm needs to take in account every one of them to succeed in the total task. Meaning it's easy to handcraft an algorithm that optimize for one client but hard to generalize.
> 3. The interesting part is in the range of <12 Mb/s for our video, over than that it becomes a trivial task - always send the max quality with zero time rebufferring.

---

> > ### Comment · Reviewer_9tk1 · 2023-11-22
> > **Thank you for comment**
> >
> > Thanks for the comments. I am still not convinced about the contribution. In particular, what is the main driving force of the algorithm which is giving it the edge over another RL-based approach. Besides, there are approaches to improve Pensieve on Puffer dataset as shown in this paper
> > https://arxiv.org/pdf/2302.12403.pdf
> >
> > Given this, I am still rejecting this paper.

---

### Official Review · Reviewer_yz9h · 2023-10-30

**Soundness:** 3 good
**Presentation:** 3 good
**Contribution:** 2 fair
**Rating:** 5
**Confidence:** 4

**Summary:**

The paper introduces a rate adaptation algorithm designed for ABR video streaming. This algorithm relies on deep-Q learning, where clients acquire knowledge about the optimal policy through training across various scenarios and numbers of video chunks. The Quality of Experience (QoE) metric under consideration is a composite of three factors: (i) chunk quality, (ii) jitter, and (iii) stall duration, which is a metric frequently examined in related studies. The authors demonstrate that their algorithm surpasses certain baseline approaches, including MPC and RobustMPC, in terms of performance.

**Strengths:**

The proposed algorithm show better performance compared to the considered baselines in the test environment.

**Weaknesses:**

The motivation and the advantage of the proposed algorithm compared to existing reinforcement learning based approaches is not so clear to me. The authors stated that:
“MPC optimization occurs online, and so setting the look-ahead window to be too far into the future can be associated with prohibitively expensive computational burden and time. “
This statement doesn't align with my perspective. To illustrate, in Pensieve (Sigcomm 2017), the look-ahead window comprises only a single predicted or estimated input to the RL agent, which serves as an approximation for solving the optimization problem. Furthermore, in practical terms, it's essential to keep the window size relatively small. As the window size increases, the accuracy of predictions tends to decrease.

Moreover, they stated that:
“In this study, we revisit ABR policy learning. We present a simple ABR scheme, called “Wolfi”2. Wolfi applies deep-Q-learning to the exact same inputs required for MPC’s online optimization process, thus replacing MPC optimization with policy learning. “
Yet, this is precisely the approach adopted in Pensieve (Sigcomm 2017): approximating the solution to the online MPC optimization problem using a reinforcement learning agent. The question then arises, how does this current work distinguish itself from the research conducted five years ago?

The authors have also put forth the following claim:
“Solve an exact optimization problem to determine the sequence of bitrates that maximizes a QoE objective (of the above form) with respect to the predicted download times.”
However, I contend that, in most cases, it is not necessary to find a solution to the precise optimization problem. Numerous algorithms have tackled simplified variants of the original problem or introduced heuristics that have demonstrated strong performance. In fact, in FastScan (published in IEEE Transactions in Circuits and Systems for Video Technology, 2019), a low-complexity algorithm was introduced to solve the exact problem. The formulation they utilized is identical to the one employed in this paper.

The issue outlined in the context of Pensieve, and I am quoting the lines here:

“However, while Mao et al. (2017) demonstrated better performance than previous methods, subsequent large- scale empirical investigation demonstrated that, when unleashed in the real world, Pensieve exhibits poor QoE (Yan et al.). This can be attributed to the gap between Pensieve’s training environment (a simulator) and the complexity of real-world traffic conditions, as well as to DRL’s notoriously brittle optimization process.”
I believe this issue is not exclusive to Pensieve itself; rather, it applies to any algorithm. In other words, any algorithm lacking training on a comprehensive dataset that accurately represents real-world scenarios will likely underperform. This is not a problem specific to Pensieve.

**Questions:**

Kindly read through the weaknesses I have outlined above.

---

> ### Author Response · Authors · 2023-11-20
>
> Thanks for your review,
> 1. below
> 2. Pensieve trained a model E2E, meaining it accepts the network condition and output the next bitrates for future chunks. Pensieve fails to perform well - Puffer show it in their paper. We divided the problem into two small problems. One, estimatet the only unknown parameter (transmission time) using Fugu and our contribution - A model that takes in account the state + the result form Fugu. The novely is our model can adapt its predicitons accordingly to the accuracy of Fugu (while MPC does not do it). Moreover our model (Wolfi) learns a policy - meaning we can replace the TTP (Fugu) but use the same policy (Wolfi) and still succeed.
> 3. I don't understand the question
> 4. Training Pensieve would take enormous time (Puffer mentioned it in their paper) But training Wolfi would take much less because we learn a generalized policy that can be applied with different models of Fugu. Learning a policy is much more stable than learning to predict on the raw inputs of network. The variance in the real world is too big to be applicatable. We show out method can be learn in a reasonable time and can be generalized to other environments.

---

### Official Review · Reviewer_GtLG · 2023-11-01

**Soundness:** 3 good
**Presentation:** 3 good
**Contribution:** 2 fair
**Rating:** 5
**Confidence:** 4

**Summary:**

The authors propose a novel Adaptive bitrate selection mechanism, called Wolfi, adaptive video streaming over HTTP. They argue that policy learning has been abandoned to optimisation based methods. The proposed method learns to adaptively change quality and provide a superior QoE with a learning based approach which is tuned using video buffers, SSIM quality of the chunks and download time over the network for the chunks.

**Strengths:**

- The paper is well written and the results are well evaluated and argued.
- Code provided was also helpful to refer from the paper.

**Weaknesses:**

- Evaluation of the streaming utilises the puffer testbed. It is mentioned that a server serves multiple clients over varying conditions using an emulator. No details are mentioned how many clients are setup. Interaction of multiple streaming clients over the network and its effects over rebuffering events, QoE, (particularly considering tcp window scaling effects in real world) is not studied.

- SSIM is not the strongest quality indicator for QoE (see question below)

-

**Questions:**

- It is not clear what was used as the quality metric for the chunk. Text says - Let Q(Ki) be the video quality at which video chunk Ki was downloaded. Video quality can be measured simply in terms of the video bitrate (as in (Mao et al., 2017)), or via more nuanced measures like the structural similarity index measure (SSIM)—a method for predicting the viewer-perceived quality of digital images and videos. What was used?

- Assuming SSIM was used to for perceived quality, VQM is a better QoE indicator and VMAF[1] can be used to compute QoE for chunks. Since QoE is one of key findings I would want to see a more robust metric over SSIM.

- Not exactly weakness but you probably don't want to write about Zoom conversation in the context of HTTP streaming as real time streaming is typically over webRTC and has nothing to do with HTTP streaming.

[1] https://github.com/Netflix/vmaf

---

> ### Author Response · Authors · 2023-11-20
>
> Thanks for your review,
> 1. We used the ssim method
> 2. Since there are many metrics we chose to use ssim because this is the method Fugu used in their paper and to our understanding is a common method. Moreover, our method does not trained for a specific metric we just optimize a function of QoE. But we can definitely add this one too.

---

### Official Review · Reviewer_NkJU · 2023-11-01

**Soundness:** 3 good
**Presentation:** 3 good
**Contribution:** 2 fair
**Rating:** 3
**Confidence:** 5

**Summary:**

The goal of this work is to present an approach for adaptive bitrate selection (ABR) in a video streaming application.  The method used is a combination of supervised learning to (i) predict the download times of different chunks followed by (ii) reinforcement leaning to decide which ones to choose.  It thus adopts the structure of a recent, successful approach that has the same prediction step, but used MPC for the second step.  This work essentially replaces MPC with an RL portion.

**Strengths:**

Well presented paper, with a clear use case and good empirical evaluation.

**Weaknesses:**

The level of contribution, particularly on the ML side is very low.  No new methods or algorithms are developed.  Essentially, the main idea is that an RL approach (using a DQN variant) may be used to produce a similar performance as an MPC in the case of video ABR.  The supervised learning part to generate some of the features (state information) used by the RL is obtained from a  previous work.  While this is a nice combination, the notion of RL as yielding a policy comparable (or superior) to MPC is well known.  This paper should probably be submitted to a networking venue such as those where Mao-2017 and Yan-2020 appeared.

**Questions:**

I don't have any questions.  I am very familiar with the past literature in this space and it is clear to me what the method does.  But I don't think this is an ICLR paper, since the contribution on the ML side is really low.

---

### Meta-Review · Area_Chair_7Uwb · 2023-11-22

**Metareview:**

The paper presents a deep-Q learning-based rate adaptation algorithm for ABR video streaming. Utilizing training across diverse scenarios and video chunk numbers, clients acquire knowledge of the optimal policy. The Quality of Experience (QoE) metric considered combines chunk quality, jitter, and stall duration. The algorithm outperforms baseline approaches, including MPC and RobustMPC, showcasing improved performance.

Strengths: results are well evaluated and argued, Code provided was also helpful to refer from the paper.

Weaknesses:
The paper is a domain paper, and does not provide any new methodology for ML domain. Even in the domain, multiple RL based algorithms have been used, and thus technical novelty seem limited.

**Justification For Why Not Higher Score:**

All the reviews exhibit a consistent perspective.

**Justification For Why Not Lower Score:**

N/A

---

### Decision · Program_Chairs · 2024-01-16

Reject